# Towards Context-Aware Facial Emotion Reaction Database for Dyadic Interaction Settings

**DOI:** 10.3390/s23010458

**Published:** 2023-01-01

**Authors:** Abdallah Hussein Sham, Amna Khan, David Lamas, Pia Tikka, Gholamreza Anbarjafari

**Affiliations:** 1Digital Technology Institute, Tallinn University, Narva mnt 25, 10120 Tallinn, Estonia; 2Enactive Virtuality Lab, Tallinn University, Narva mnt 25, 10120 Tallinn, Estonia; 3iCV Lab, University of Tartu, Narva mnt 18, 51009 Tartu, Estonia; 4iVCV OÜ, 51009 Tartu, Estonia; 5PwC Advisory, 00180 Helsinki, Finland; 6Institute of Higher Education, Yildiz Technical University, Beşiktaş, Istanbul 34349, Turkey

**Keywords:** facial expression analysis, emotion recognition, emotion reaction, responsible AI, data collection

## Abstract

Emotion recognition is a significant issue in many sectors that use human emotion reactions as communication for marketing, technological equipment, or human–robot interaction. The realistic facial behavior of social robots and artificial agents is still a challenge, limiting their emotional credibility in dyadic face-to-face situations with humans. One obstacle is the lack of appropriate training data on how humans typically interact in such settings. This article focused on collecting the facial behavior of 60 participants to create *a new type of dyadic emotion reaction database*. For this purpose, we propose a methodology that automatically captures the facial expressions of participants via webcam while they are engaged with other people (facial videos) in emotionally primed contexts. The data were then analyzed using three different Facial Expression Analysis (FEA) tools: iMotions, the Mini-Xception model, and the Py-Feat FEA toolkit. Although the emotion reactions were reported as genuine, the comparative analysis between the aforementioned models could not agree with a single emotion reaction prediction. Based on this result, a more-robust and -effective model for emotion reaction prediction is needed. The relevance of this work for human–computer interaction studies lies in its novel approach to developing adaptive behaviors for synthetic human-like beings (virtual or robotic), allowing them to simulate human facial interaction behavior in contextually varying dyadic situations with humans. This article should be useful for researchers using human emotion analysis while deciding on a suitable methodology to collect facial expression reactions in a dyadic setting.

## 1. Introduction

In the near future, social activities between people are envisioned to expand into immersive virtual environments, even replacing many physical world situations. These environments will also be inhabited in an increasing manner by artificial humans, such as virtual service personnel or social accompaniment. Creating contextually relevant facial behaviors for artificial humans, however, continues to be a challenge. One of the key tasks for overcoming this challenge is the creation of dynamical databases that would allow artificial characters to “learn” variations of human-to-human interactions in socially meaningful situations. The goal of our work is to contribute to this task by developing a method for collecting contextually conditioned dyadic emotion reaction databases that could be used in creating socially credible virtual humans.

Artificial humans, be these social robots or virtual avatars, may react in a contextually relevant way to observe human facial expressions only if the machine learning models that drive the behavior have been trained with such data. In this paper, we describe our data collection method in which we recorded participants’ facial behaviors on video when they were viewing a set of stimulus videos containing natural facial expressions of different people. To simulate social variations of facial encounters, the viewing of stimulus videos was primed with varying psychological contexts. The aim of inducing social simulation in the participant’s engagement task was to increase the number of dynamical facial variations in the tracking condition.

Facial Expression Recognition (FER) has gained popularity over the past decade in computer vision and affective computing. The challenge to recognizing facial emotion is important as it is believed that two-thirds of communication is conveyed through nonverbal communication, whereby facial expressions are one of the main channels [1]. Emotions and their physiological expressions are assumed to be the most-significant indicators, as they reflect social roles and convey knowledge of people’s thoughts, expectations, and social experiences [2]. Although facial expressions and gestures vary from human to human, it is important to recognize emotion as, in some cases, it can be very crucial. G.A. Van Kleef (2009) proposed a new framework: the Emotional as Social Information (EASI) model. This framework helps in understanding the interpersonal effects of emotions. It focuses on discrete emotions, rather than diffuse moods, and distinguishes two processes by which emotional expressions influence interpersonal relationships [3]. For instance, the EASI model can help in studying the interpersonal effects of emotions on behavior in parent–child interactions [3] or whether anger can enhance creativity [4]. On the other hand, people diagnosed with autism show difficulty recognizing facial expressions (see the review [5]), and having an automatic FER system at their disposal can be helpful.

For the realization of databases that contain dynamic material on how humans react to others in human–human dyadic interactions, large amounts of interaction data need to be collected. This endeavor includes contextually relevant metadata, i.e., automated and manual annotations synchronized with the dynamically changing behavioral data. Typically, databases that are used for the training, testing, and validation of emotion recognition algorithms rely mainly on discrete emotion labels or on a continuous arousal–valence scale. In controlled data collection conditions, bodily behaviors can be tracked based on psycho-physiological measures. Well-established methods include Electroencephalography (EEG), Electrocardiography (ECG), Galvanic Skin Response (GSR), Skin Temperature measurements (SKT), Electromyography (EMG), Electrooculography (EOG), as well as Facial Expressions (FEs), Body Posture (BP), and Gesture Analysis (GA) (for a review, see [6,7,8,9]). The downside of many of these methods—regarding the freely unfolding social interaction—are the sensors themselves, which need to be attached to the body of the participant, thus potentially inducing distracting elements to the study condition.

All behavioral laboratory studies are, to some extent, inclined to the so-called Hawthorne effect, the tendency of participating individuals to change their behavior due to awareness of being observed [10]. This effect can be partly reduced by innovative experimental designs. For example, the high-resolution webcams embedded in stimulus monitors may not draw participants’ attention in a similar manner as larger cameras, thus allowing them to capture facial expressions from users in a less-intrusive way. Such a setting was used in gaming studies [11], where players’ facial expressions during a simple arcade-style computer game were recorded using a webcam and analyzed in the Affectiva Affdex SDK platform, which uses deep learning (for details, see [11]).

Previous work, such as [12,13,14,15,16,17,18], has used deep learning models like the Convolutional Neural Network (CNN) for emotional expression recognition with relatively high accuracy. However, so far, only a few studies that use available machine learning methods have investigated dyadic facial expression reactions specifically targeted for Human–Robot Interaction (HRI). One of the challenges is how to annotate the rich bodily behaviors taking place in the dyadic interactions. As an example to overcome this challenge, a multimodal corpus RECOLA was created, which contains not only psychophysiological data collection (ECG, EDA), but also arousal, valence, and social behavior annotations of audiovisual recordings where people interact in dyads [19]. Several projects have addressed the data collection and annotation of human affect behaviors in naturalistic contexts, such as HUMAINE [20], SEWA [21], or the Aff-Wild Database [22]. While the focus of the current paper is on the dyadic face-to-face interaction, however, similar issues are related to the human–robot interaction studies that go beyond dyads towards multiagent (non-dyadic) interactions [23].

When the recorded data of the participant’s bodily behaviors are linked with the subjective experiences reported by the participant, the collected data can be better annotated and validated, also allowing a more in-depth interpretation of the dyadic situations. For example, the emotional database by T. Sapiński et al. [6] in which facial expressions, bodily gestures, and speech performances of professional actors were annotated using Ekman’s emotion categories was validated by a group of experts and volunteers. On a different note, in [24], a multi-subject affective database on participants’ facial expressions included recorded physiological signals (oxygen saturation, GSR, ECG) linked to the viewing of emotional film clips, was validated based on the participants’ self-reports about the felt emotions.

A range of methods allow synchronization, filtering, segmenting, and automated classification of the raw data collected from human participants [25,26,27,28]. For example, in [6,24,29], the data capture was all synchronized when triggered, a challenge in its own right. For instance, the participants’ reactions need to be aligned with what they are seeing. Regarding how sensitive humans are to noticing discrepancies in human facial expressions even in milliseconds, any delay in facial data synchronization may induce distortions to the dataset. The work presented in this paper can be seen in line with the recent studies on dyadic human–robot interaction, for instance [30,31,32,33].

Our contributions to this study field are as follows:We describe a novel method of collecting participants’ facial expressions while watching videos of different emotional expressions of other people.We develop an algorithm that automatically collects the dynamical facial emotion reaction database for dyadic interaction settings and automates the whole process to limit human intervention during the data collection/experiment.We pre-processed the database and analyzed it using three different models as a comparative analysis.

We organize the paper in the following structure. The experimental setup is described in Section 2. The data acquisition is in Section 3, where we explain each step concerning the algorithm, experimental procedure, settings, and questionnaires. The results and discussion are in Section 4, where we provided a thematic analysis of the responses we received from the participants through questionnaires and the results from the three different models. We conclude by outlining future work based on the collected dyadic emotion reaction data in Section 5.

## 2. Experiment Overview

The experiment was approved by the Ethical Committee of Tallinn University and conducted in a controlled lab setting.

### 2.1. Participants

Sixty healthy volunteers (thirty females, thirty males) ranging from 18 to 59 years old were provided with complete information about the research’s experimental nature, contents, and methodology. They signed an informed consent form before the experiment.

The volunteers were contacted via the university email lists, social media (mainly Facebook), and personal networks. The experiment was conducted in English, and all information was provided in English. Thus, good English language skills were required to participate. The participants were offered an EUR 10 gift card as a token of appreciation at the end of the experiment.

By “participant”, we refer to the person who takes part in the experiment in the lab. By “subject”, we systematically refer to the person whose behavior was pre-recorded in the database videos that the participant would be engaging with in the lab experiment.

### 2.2. Motivation for Social Contexts

In our study, we specifically aimed to create experimental conditions that would simulate a socially conditioned dyadic facial encounter. During the experiment, we collected data on dynamical behavioral patterns in contextually primed face-to-face situations. Each participant’s facial expression reactions were recorded while watching a series of recorded videos of different people expressing emotions labeled neutral, happy, unhappy, angry, and surprised. We primed the participants with five contextual social situations so that we could obtain different variations of facial reactions. Following this strategy, the people in the videos were introduced as “the best friend”, “colleague”, “stranger”, “a person you hate”, or “someone you are afraid of” (five different scenarios). Then, they were asked to respond to the person’s expressions in the video as they would respond in that specific setting in real life.

After the experiment, we provided a questionnaire to each participant to collect qualitative data on the participants’ subjective experiences related to the felt naturalness of the socially contextualized situations and the experimental procedure in general. The responses were given in textual form.

## 3. Data Acquisition

### 3.1. Experimental Procedures

Each participant was prepared for the experiment, provided general instructions and the task description, signed the consent form, and provided a training session for the task, as in Figure 1. Once the participant was ready, we started the recording session, which would last about 30 min.

The experiment had five different primed social situation iterations. In each iteration, we dedicated 10–12 s to display the socially contextualized scenario (for example, “Suppose these people are your best friends”). Each iteration lasted around 4 min, during which the participant saw 3 different male and 3 different female subjects expressing different facial expressions. After each iteration, there was a 2–3 min break before the next iteration started. After the fifth scenario (last iteration), the participants were asked to fill out the questionnaire.

Figure 2 shows the laboratory setting. It was organized so that any potential distraction between the participant and the experiment monitor was removed. In the room, cables were hidden from the field of vision, and the participant was secluded with cardboard. A white noise generator was used to blend with the outside noise using speakers.

### 3.2. Hardware Materials

The following materials were used as the hardware:A Logitech webcam HD 1080p.A computer for running python scripts and editing and storing videos. We used a Lenovo ThinkPad L480 (i7 core CPU, 16 GB with no graphic cards support).Hard drive for video storage.Monitor/TV screen displaying the pre-recorded videos.Speakers for white noise.

### 3.3. Video Materials

We randomly selected six male and six female subject videos from the iCV lab’s “Sequential Emotion Database” [27]. The subjects in the database performed different facial expressions in two distinct sequences as follows:NHUAS: Neutral, Happy, Unhappy, Angry, Surprise (Sequence 1);NAHSU: Neutral, Angry, Happy, Surprise, Unhappy (Sequence 2).

We propose that the participant watch the same subject twice (both sequences) in each iteration. Therefore, we made some amendments. First, we divided the number of subjects equally for the trial and actual experiment, as shown in Figure 3. From the selected six different male and female subjects, we divided them into two sets, the first one with three pairs of males and females for the training/trial part and the other three pairs in the actual experiment.

Now that we had pairs of male and female subjects, we could alter the positioning of the three pairs to prevent participants from predicting future subjects and becoming used to the organization of the subjects in the videos. These videos were also arranged to have different pairs of combinations (Female–Male (F-M) and Male–Female (M-F)) subjects. We used the F-M and M-F patterns for the subjects so that the end results had the following combinations with the participants: M-M, M-F, F-M, and F-F. In other words, the subjects were designed to be in alternating combinations of male and female subjects, while the participants were to vary sixty times in our gathered database.

To clarify the ordering, consider the following: If we would use a female–male gender pair, the first gender would be Female 1–Male 1, Female 2–Male 2, and Female 3–Male 3. The said order can be formulated as Variation 1: Pair 1, Pair 2, Pair 3. We can then shuffle the pairs in a different order, as shown in Table 1. One of these variations was chosen twice in one iteration during the trial and actual experiment at random by our algorithm (see Section 3.4), and participants would see two different groupings as tabulated in Table 2 in the same iteration where they would see Group 1 and, then, Group 2.

Since we were already shuffling the variations twice in each iteration, we decided to use Sequence 1 in the first half and Sequence 2 in the second half. In this way, we could compare how differently the participants would react. Here is an example of how it worked during both the trial and actual experiment to better understand it.

Suppose Participant X signed the informed consent form and is ready to perform the trial. Before beginning the experiment, X is reminded again of the task with the following content: “You will be seeing videos of different people facing you. Respond to their expressions as you would respond in a real-life situation.”:
X will see the first scenario displayed on the monitor as shown in Figure 4B for 10 s, then X will start seeing the subjects from the newly edited database.X will first see a female subject, then a male of the same pair, say Pair 2.Then, X sees another female, then a male subject from Pair 3.Lastly, the final Pair 1 is seen.Then, X would see male–female subjects from Pair 1.Afterward, Male and Female Pair 2, and finally Male and Female 3 would be seen.**Note:** For now, the participant has seen the subject exhibiting emotions based on Sequence 1, and when the order of the gender changes, the sequence changes as well to Sequence 2 in the same scenario/context with two different variations.Once the above procedure is complete, our algorithm (see Section 3.4) compiles the videos and displays the next scenario/context.Go to Step 2 (loop). The same procedure would repeat for all the other scenarios/contexts (mentioned in the next point). However, the variations in Steps 2 to 6 would be changed by our algorithm. Hence, this would summarize the time graph in Figure 1.**Note:** For the trial part, the algorithm (see Section 3.4) would stop at Step 7 with different subjects, as explained earlier, and we used other different subjects for the actual experiment.

**Figure 4 sensors-23-00458-f004:**
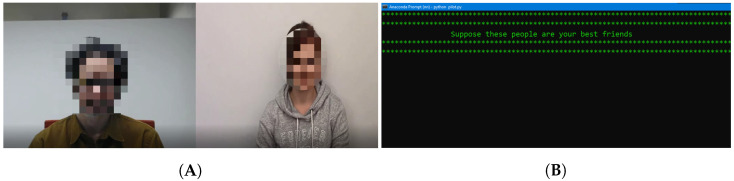
From (**A**), on the left, a screenshot of the actual result after saving the compiled video. From (**B**), on the right, as soon as the trial or the experiment starts, the participant will see this on the monitor (Scenario 1).

Before beginning the actual experiment, the participant is told that the subjects in the trial and actual experiments are different. It is also important to highlight that, before the participants saw each subject, we prepped them with a 5-s countdown on a black screen. The scenarios were displayed to the participants as follows:Scenario 1: Suppose these people are your best friends.Scenario 2: Suppose these people are your acquaintances or colleagues.Scenario 3: Suppose these people are strangers.Scenario 4: Suppose these people are people you hate.Scenario 5: Suppose these people are dangerous and you are afraid of them.

### 3.4. Algorithm

Since we did not want to distract the participant during the experiment, it was important that the videos were shown to the participant smoothly without any human involvement. Therefore, we developed an algorithm that would automate the whole process using the python programming language and the following libraries: *NumPy*, *moviepy*, *OpenCV*, *imutils*, and *random* (https://github.com/ahsham/Dyadic-Reaction-Emotion-Database, accessed on 27 December 2022).

The main challenge was to save the two different, but interrelated video files in a synchronized manner: the video recording of the participant and the set of corresponding stimulus videos. We achieved this by recording the screen while the video was being played and recording the webcam feed simultaneously. We then resized each video to 720p in the MP4 format and placed them side-by-side using OpenCV, as shown in Figure 4A, and when participants saw the screen, as displayed in Figure 4B.

Whenever a new entry was made, we conditioned the algorithm to check whether a similar file name already existed. If it did, we prompted the researcher to check if the file name needed to be altered or not. Else, we over-wrote it. We compile the rest of the algorithms in Figure 5 and Figure 6.

As explained earlier with Participant X, in each iteration, there are two different sequences, and Figure 6 shows how the videos were processed and compiled. Figure 5 shows how our algorithm checks if the researcher is performing a trial or an actual experiment. The algorithm executes only one of the scenarios for the trial, but all five for the actual experiment.

Once the participant had completed all the aforementioned scenarios, the experiment was over. We then handed his/her a questionnaire, which comprised seven questions. The first two questions provided information on the participant’s performance, that is if he/she felt the experiment was too long or if he/she had understood (or not) the given task properly. The next three questions related to the face-to-face simulation experience and how natural he/she felt the situation was. The sixth question was for detecting indications if he/she had become tired during the experiment. This and the very last question also sought to investigate the modifications to the experiment recommended by the participants.

The questions were as follows:How did you find the length of the experiment?Was the information about the experiment provided to you helpful and delivered in a clear manner?Did you find the people in the videos relatable?Was it difficult to respond to videos as if you were in a real-life situation?Did you feel your responses were natural?Did you feel that you wanted to have an additional break during the experiment?Do you have any final remarks or suggestions?

## 4. Results and Discussion

The study aimed to capture nonverbal facial expressions to explore dynamic behavioral patterns in human faces when engaged in face-to-face situations with other people. It included collecting the physiological facial behaviors of the participants (quantitative data) and subjective reports of their experiences (qualitative data). In this section, we first present the qualitative findings from the questionnaire and then discuss a comparative video data analysis of the three different models or Software Development Kits (SDKs).

As mentioned earlier, the HUMAINE [20], RECOLA [19], SEWA [21], and Aff-Wild [22] databases were collected with a focus on the facial expression recognition when two or more people are having conversations. Deviating from these, our research focus was on the facial emotion reaction recognition during nonverbal dyadic face-to-face interaction.

### 4.1. Questionnaire Data Analysis

All sixty participants replied to the questionnaire in textual form. To investigate more about the details of the process from non-numerical data, the data were analyzed using the qualitative data analysis software NVIVO 12.

The data were organized in the following manner: The data patterns were visualized graphically for the six queries. The responses collected for the last question reflect participants’ feedback for modifications or suggestions in the experimental procedure.

For the question of whether the information about the experiment was helpful and delivered in a clear manner, fifty-six participants agreed that it was clear and helpful, while four reported it as vague or unclear.

The question related to the length of the experiment, which was in total 45 min long. Forty-seven participants reported feeling comfortable about the length of the study, while seven found it lengthy, four reported it as short, and two did not provide an answer. For the sixth question, related to an extra break, forty-seven participants reported that they did not need an extra break, while the remaining participants proposed to add a break during the experiment to increase cognitive productivity.

Based on the post-experiment questioning, the majority of them stated that they felt their responses were natural, as if in real situations.

During the experiment, the video showed individuals with different emotional expressions. For validating if the collected facial data were under the conditions of the participants simulating social situations as if they were taking place in real life, Questions 3–5 were most relevant. Thirty-nine reported that they felt like responding to the relevant people because of familiar expressions, while ten reflected them as not relatable, and eleven were undecided.

Reaction to the videos was critical to determine whether they felt the same way as they would when interacting with people in real-life situations. The experiment was designed to make the participants feel as if they responded to the emotions in contextually relevant real-life situations (e.g., simulating a friend or a person of dislike). In this regard, forty-six participants stated that their own facial expressions toward the people in the video were natural; this is similar to as if being in real-life situations. Sixteen believed it simpler to respond in real life rather than in videos, for instance: “Yes, I imagine that I would respond more expressively in real-life situations”.

The responses of participants showed variations according to different age groups. Participants under the age of 20 tended to be more at ease and interested in participating in the experiment because they found the characters in the video familiar, and they seemed to be more creative about being in real-life circumstances. The majority of the participants in the age group 20–30 years were satisfied with the length of the trial and declared the experimental knowledge reasonable, requiring no further breaks. Most people in this age group found the feelings of the characters in the video familiar, but there were differences in how they imagined themselves in real-life scenarios. Thirteen participants between the ages of 30 and 40 years stated that the duration of the experiment was acceptable or sufficient, but they did not find the people in the videos to be relatable, making it challenging to imagine interacting in real-life scenarios. People in the age group of 40–50 years could relate to the people in the videos, but they were unsure if their response toward them was natural. They did not find it challenging to respond as if in real-life scenarios. People over 50 years of age reported that they were unable to react promptly to the videos and suggested that the videos be slowed down.

Compared to males, women declared their responses to be more natural as if they were in real-life circumstances. Sixteen out of sixty participants (primarily men) suggested improvements. The majority centered on adding additional individuals for different circumstances, such as for friends; in other words, the people presenting expressions as friends in the video should be distinct from those represented in the group of enemies.

#### Truthfulness Validation

We checked the truthfulness of the participants’ emotions using 4 different individuals (2 males and 2 females) aged between 30 and 60 years. They all confirmed that these emotions were true, and they were all not the same, in the sense that the participants were not mimicking the subjects and they were actually reacting genuinely. Younger participants seemed to react faster and to be more focused than older ones. The female participants seemed to be more expressive than the male ones.

### 4.2. Video Data Analysis

We pre-processed the videos for further analysis using three different models: iMotions, Mini-Xception model (from our previous publication), and Py-feat FEA toolkit.

#### 4.2.1. iMotions

We processed our dyadic videos using the software iMotions [34], a commercially available API that can predict facial expressions using the Affectiva algorithm. The limitation of the software turned out to be that it can only process one Facial Expression Analysis (FEA) at a time. To overcome this, we amended the videos accordingly so that we could have the analysis for both the participant (on the left in the video setting) and the subject (on the right in the video setting).

Figure 7 shows the overall emotions of the subject, and it is significantly co-related to the ground truth, where the subject starts with a neutral facial expression, then happy, unhappy, angry, and surprised. When we repeated the same for the participant, we obtained the following result as shown in Figure 8. When the result was compared manually with the video, iMotions’ prediction was a bit aligned with the actual reactions.

#### 4.2.2. Py-Feat FEA Toolkit

The free Py-Feat FEA toolkit is suitable for analyzing facial expressions for both images and videos [35]. In comparison to iMotions, the authors suggested that their software is slightly more accurate. In our analysis, the results for the same subject were as follows:

From Figure 9, the result for the subject is not co-related to either the ground truth or to the iMotions results. The result for the participant is as follows.

From Figure 10, the happy emotion had a similar correlation to iMotions, but the fear emotion was completely out of range. It looks like the model’s prediction needs to be normalized.

#### 4.2.3. Mini-Xception Model

Based on the previous study [33], we next applied the Facial Expression Analysis (FEA) model that had worked well for videos over a huge number of frames (more precisely, over 2000 frames). We used this model to compare the above two models. The authors from [33] did not use discrete basic emotions, but instead selected the valence scale from the continuous arousal–valence scale. One of the reasons why it was found useful to apply only the valence scale can be understood from Figure 11. In this case, emotion is more like a regression problem. The overall valence of the subject is shown in Figure 12.

From Figure 12, the result for the subject was more or less co-related to the ground truth, but not to the other two results. We then retrieved all the valence percentages to have a clearer picture, as shown in Figure 13.

From Figure 13, the result for the subject was more or less co-related to the ground truth, but not to the other two results. The result for the participant was as depicted in Figure 14.

### 4.3. Summary

iMotions software can only process one person’s facial expression from videos. Due to this limitation, we used iMotions separately for tracking the facial expression of the subject videos and then on the participant videos. To exemplify our findings, including the other two models (Py-Feat, Mini-Xception), here, we summarize the dyadic results separately for only one subject’s facial emotion behavior and one participant’s facial emotion reaction behavior, respectively.

Subject analysis: As mentioned earlier, we ran the analysis accordingly and obtained Figure 7. First, we compared the results of Figure 7 with our ground truth, which was taken from the dataset. The ground truth was in the following sequence: neutral, happy, unhappy, angry, and surprised. Figure 7 shows similar results, which are more or less related to the ground truth. Comparing the same video of the same person using the Py-Feat FEA toolkit, we did not see the same results in Figure 9. Instead, we saw anger as the maximum, and the overall spectrum of the emotions did not correspond exactly to iMotions. When compared with the Mini-Xception model in Figure 13, we saw that there was a variation of positive, neutral, and negative valence. The results show that the valences were in the following order: neutral, positive, neutral, negative, and positive valences. The ground truth, when compared to the valence scale, was as follows: neutral, positive (happy), negative (unhappy), negative (angry), and positive (surprised). Here, we can see only the unhappy emotion, which was not in line with the ground truth.

Participant analysis: In contrast to subject analysis, when we compared the results for the participant, we saw that the person reacted with a happy facial expression while the stimulus smiled and acted surprised. After confirming this fact with our participants (who validated the truthfulness), the results were aligned with Figure 8. When we look at Figure 10, the Py-Feat FEA toolkit did not produce exactly the same prediction. To our best knowledge, we think why we saw such a result is probably that the model was not properly normalized. Finally, when we look at the results in Figure 14, we see that the model predicted that the person reacted with a neutral facial expression. Again, we think that the model needs to be tweaked for emotion reactions.

The quantitative findings from comparing the three existing models showed that, while they could predict the facial expressions in the videos relatively accurately, they were not accurate for predicting the participants’ emotion reactions related to the observed expressions of the subjects in the stimulus videos. This finding calls for further exploration.

## 5. Conclusions

Emotion identification and evaluation play a vital role in creating various human–machine interaction systems. Facial emotion recognition is a robust method that allows the machine to assess dynamically changing emotional states and predict the future behaviors of the human counterpart. This is crucial for developing meaningful interaction functions of the machine, so that it may accordingly deliver the most-appropriate feedback to the human. In our view, the best way for the machine to learn such interactions is to learn them from the human-to-human encounters. Although the relationship between specific perceived emotions and human reactions has long been established, there are numerous ambiguities in choosing appropriate methods for collecting facial expression reactions in a nonverbal human-to-human dyadic setting. In this paper, we proposed a non-intrusive method to collect the dyadic emotion reaction data of participants watching strangers expressing different facial expressions on video in different contexts.

We developed a python script that can collect dyadic facial emotion reaction data elicited in the participants when they viewed other people’s facial expressions in the video recordings. The viewings were primed with different socially relevant contexts, where the participant was asked to simulate the subject in the video, for example as a friend or as a stranger. After the experiment, we validated the naturalness of the participant responses (genuine emotion) to the subject’s expressions in the video with other volunteers. From the qualitative reports, the described experimental design seemed to provide an adequately relevant simulation of human-to-human encounters in natural contexts. Most participants described the dyadic interactions as comfortable and felt that the persons in the videos were relatable similarly to real-life situations. These results support our proposed method for collecting a dyadic emotion reaction database. The results from the comparative analysis showed that different facial expression recognition software, when applied to our collected data, could not agree with a single emotion reaction prediction. Based on this result, a more robust and effective model for emotion reaction prediction is needed.

We envision that the dyadic database initiated in this study can be used to further develop advanced models for identifying the facial emotion reactions of social robots and virtual humans. Such a methodology and the implementation of machine learning for data analysis appear to be an extremely powerful combination. We believe that our work will be useful for researchers using human emotion analysis while deciding on a suitable methodology to collect nonverbal facial expression reactions in a dyadic setting.

## Figures and Tables

**Figure 1 sensors-23-00458-f001:**
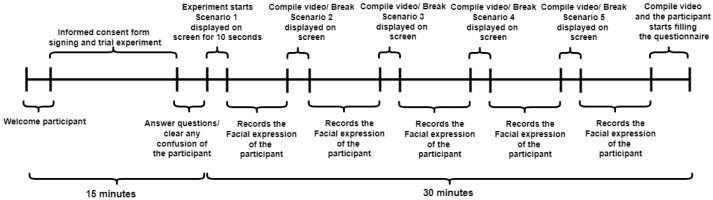
Time graph of the whole experiment.

**Figure 2 sensors-23-00458-f002:**
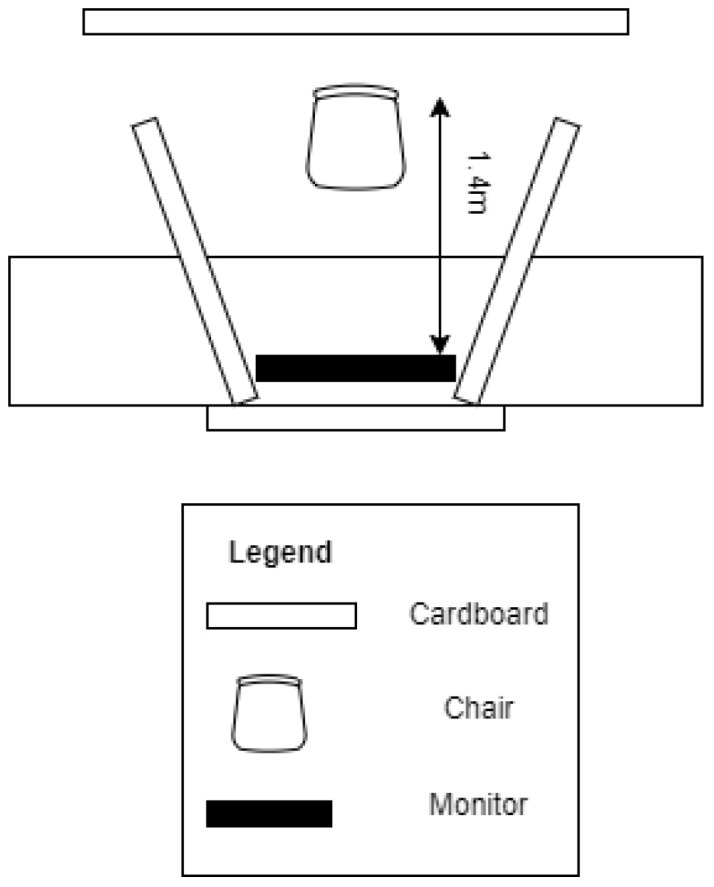
Aerial view of experimental setting during the experiment.

**Figure 3 sensors-23-00458-f003:**
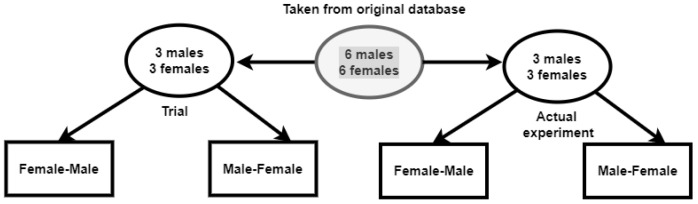
Video editing sample distribution into trial and actual experiment.

**Figure 5 sensors-23-00458-f005:**
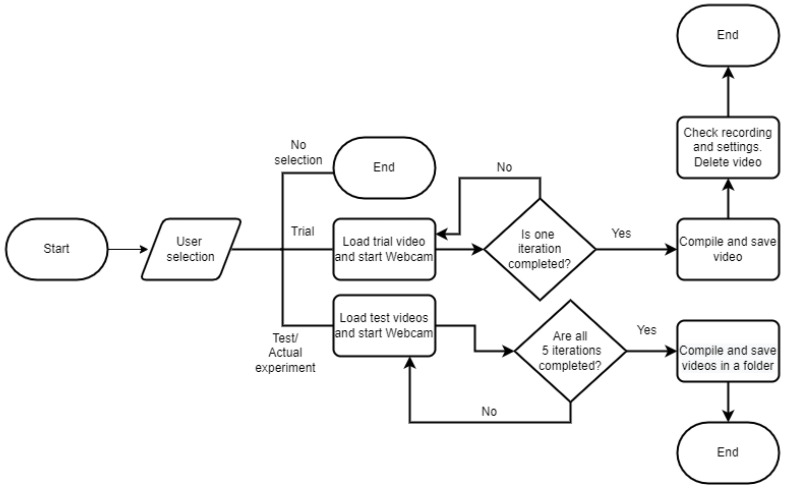
Overall description during both trial and test experiment.

**Figure 6 sensors-23-00458-f006:**
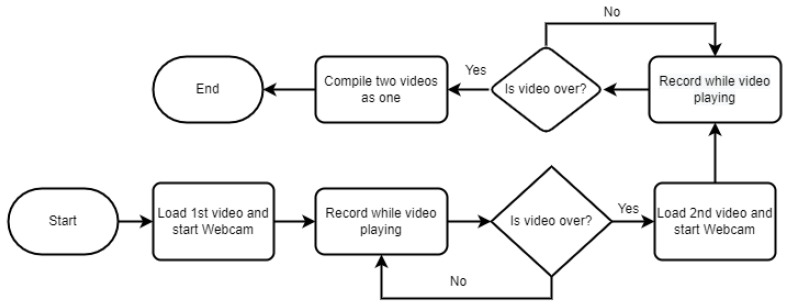
Steps during one iteration.

**Figure 7 sensors-23-00458-f007:**
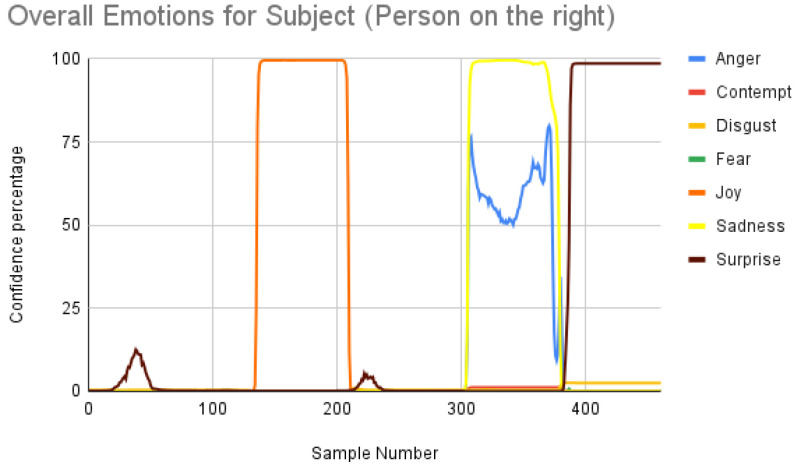
Overall emotions of the subject (the person on the right) for one random sample using the iMotions SDK.

**Figure 8 sensors-23-00458-f008:**
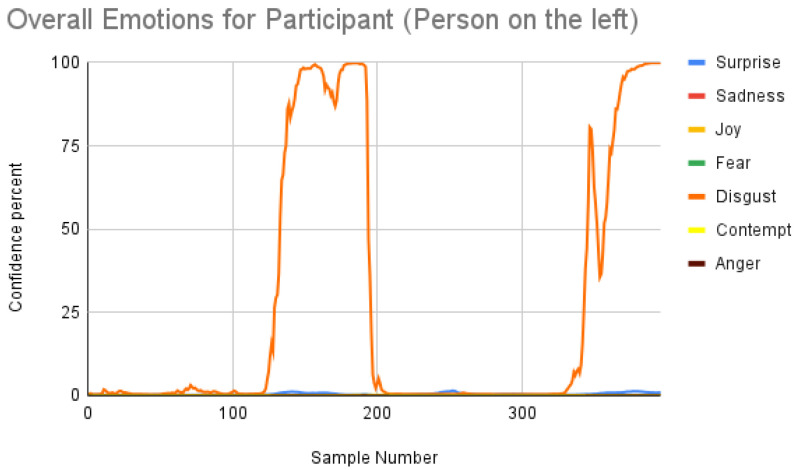
Overall emotions of the participant (the person on the left) for one random sample using the iMotions SDK.

**Figure 9 sensors-23-00458-f009:**
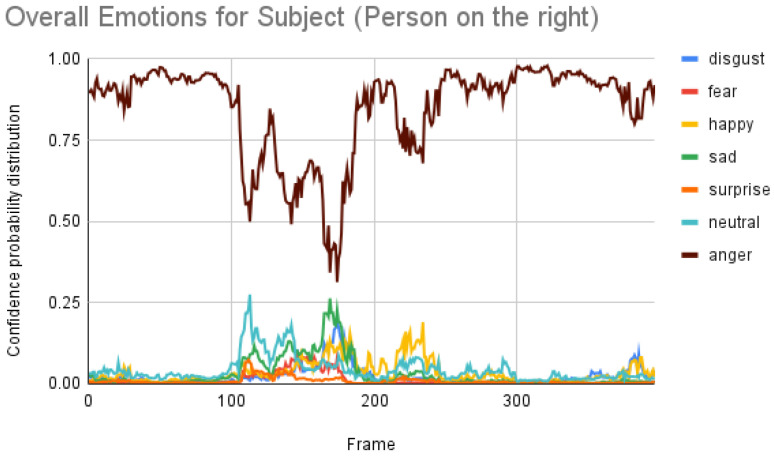
Overall emotions of the subject (the person on the right) for one random sample using the Py-Feat FEA toolkit.

**Figure 10 sensors-23-00458-f010:**
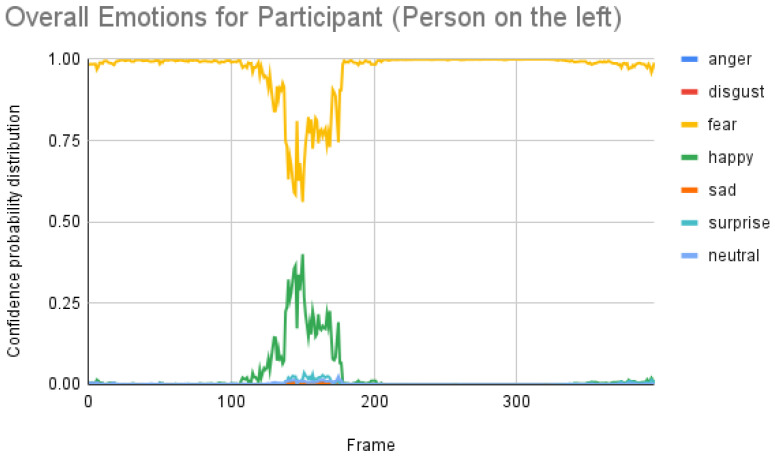
Overall emotions of the participant (the person on the left) for one random sample using the Py-Feat FEA toolkit.

**Figure 11 sensors-23-00458-f011:**
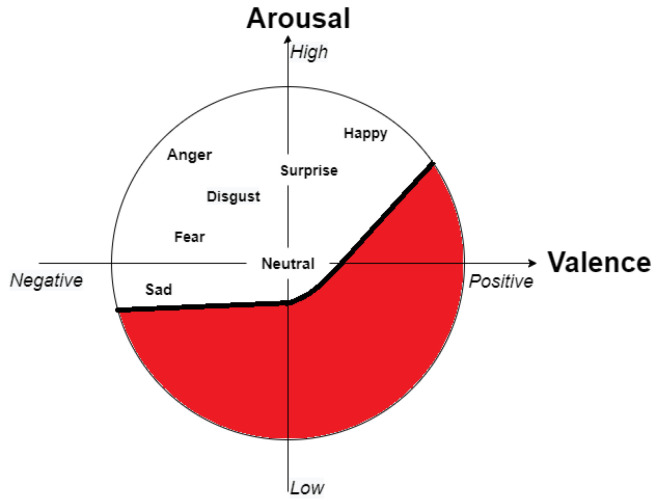
Valence–arousal scale taken from [36]; the newly modified red portion represents the section that was not used from the seven basic facial expressions.

**Figure 12 sensors-23-00458-f012:**
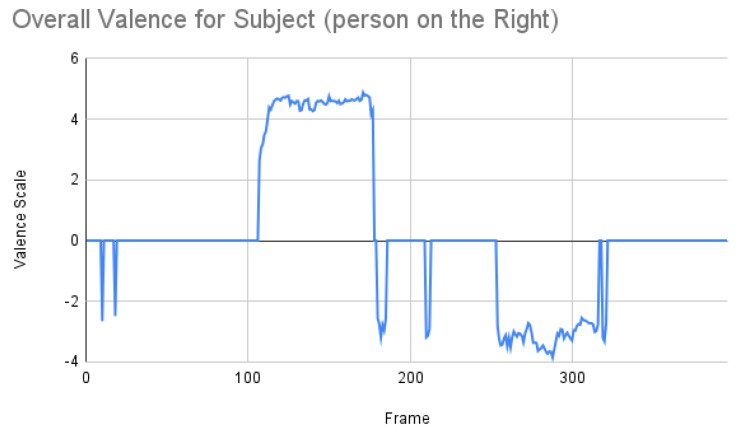
Overall emotions of the subject (the person on the right) for one random sample using the Mini-Xception model.

**Figure 13 sensors-23-00458-f013:**
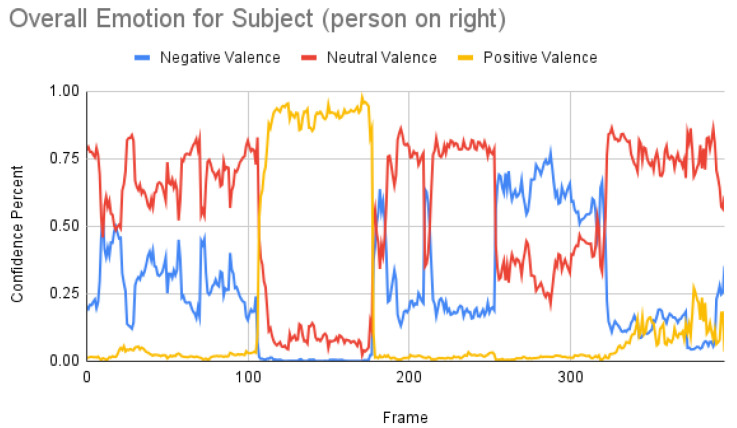
Overall emotions with all valences of the subject (the person on the right) for one random sample using the Mini-Xception model.

**Figure 14 sensors-23-00458-f014:**
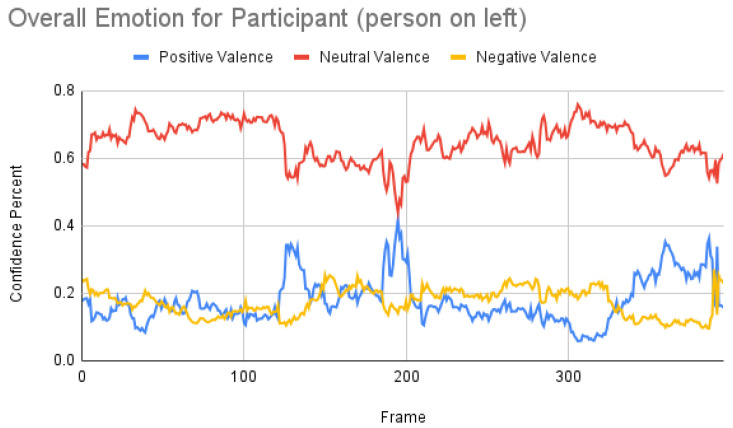
Overall emotions with all valences of the participant (the person on the left) for one random sample using the Mini-Xception model.

**Table 1 sensors-23-00458-t001:** Each pair (either male–female or female–male) is organized in different variations as highlighted in bold.

Variation	Pair Order	Pair Order	Pair Order
**1**	1	2	3
**2**	2	3	1
**3**	3	1	2
**4**	1	3	2
**5**	2	1	3
**6**	3	2	1

**Table 2 sensors-23-00458-t002:** The participants will see the videos in the two sequences in each iteration.

**Group 1 (by gender)**	woman	man	woman	man	woman	man
**Group 2 (by gender)**	man	woman	man	woman	man	woman
**Duration (in seconds)**	6 + 15	6 + 15	6 + 15	6 + 15	6 + 15	6 + 15

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
