# Peer review of "Towards Context-Aware Facial Emotion Reaction Database for Dyadic Interaction Settings"

_sensors, 2023, doi:10.3390/s23010458_

Round 1

Reviewer 1 Report

In the current study healthy adult participants responded with facial expressions to videos of people expressing various emotions. The participants were also primed with different social contexts to elicit different variations in their expressions. The authors report their findings based on qualitative analysis of the participants’ subjective descriptions of the dyadic interactions used in the experiment and a quantitative analysis of emotions expressed in the experiment using iMotions, Mini-Xception, and Py-Feat FEA toolkit. Overall, it was reassuring to see that the participants felt that watching videos was like naturalistic interactions with people in real-life situations (ecological validity). Although the models were able to predict the emotions portrayed in the videos, they were not as good in predicting the emotions of people watching them. This is an interesting and well-done study and I recommend its publication. I have a few minor suggestions which I want the authors to consider.

1. In the introduction, the authors should discuss the importance of emotion recognition.  Why is emotion recognition important? What are the consequences of not being able to recognize emotions? They can even talk about how this can affect populations such as autism spectrum disorders. Taking a look at The Emotions as Social Information (EASI) Model might be a good starting point.

2. In the discussion, the authors should make references to other similar studies done in the past. Are the findings similar or different from them?

6. Finally, the authors have to provide a detailed explanation of the discrepancy in the models’ prediction of emotions in the videos vs subjects’ reactions to the videos. I did not find any convincing explanation for this.

Author Response

We thank the reviewers for their suggestions and comments, which we found very valuable. We have made corrections and additions accordingly. Below, please find the detailed replies. 

Reviewer comment:

  1. In the introduction, the authors should discuss the importance of emotion recognition.  Why is emotion recognition important? What are the consequences of not being able to recognize emotions? They can even talk about how this can affect populations such as autism spectrum disorders. Taking a look at The Emotions as Social Information (EASI) Model might be a good starting point.

Author’s response:

Thank you for your suggestion. It is true that the importance of emotion recognition needs to be addressed. We have added a new paragraph in red on the second page in the introduction section. 

Reviewer comment:

  1. In the discussion, the authors should make references to other similar studies done in the past. Are the findings similar or different from them?

Author’s response:

We have revised the manuscript and made the correction in red.

Reviewer comment:

  1. Finally, the authors have to provide a detailed explanation of the discrepancy in the models’ prediction of emotions in the videos vs subjects’ reactions to the videos. I did not find any convincing explanation for this.

Author’s response:

We have added additional explanations as requested in the summary section 4.3. We have revised the manuscript and made the correction in red.

Reviewer 2 Report

The article was written correctly.

Clearly the authors have and experiment ongoing and a subject that may be the content for publication.

Correct spelling, punctuation, and grammar make all the difference. Always keep in mind that comments are where you engage with other scientists on a professional level.

The work in the current state must be improved, in the following points:

The presentation of the proposal must be clear and precise.

All the reference is used in the manuscript should be in order form. In this manuscript mostly the reference was used randomly.

Legend in Figures (a) and (b) is not in the same location, it should be in the same location.

In addition, the conclusions and abstract must be self-contained.

In Methodology Section

The work in the current state must be improved, in the following points:

The presentation of the proposal must be clear and precise. Including models, algorithms, and diagrams.

A further suggestion for citation, cite the following ready reference in this manuscript

  1. Ullah, R., Gani, A., Shiraz, M., Yousufzai, I.K. and Zaman, K., 2022. Auction Mechanism-Based Sectored Fractional Frequency Reuse for Irregular Geometry Multicellular Networks. Electronics11(15), p.2281.

Author Response

We thank the reviewers for their suggestions and comments, which we found very valuable. We have made corrections and additions accordingly. Below, please find the detailed replies.

Reviewer comments:

The work in the current state must be improved, in the following points:

The presentation of the proposal must be clear and precise.

Author’s response: Thank you for your comment. We have revised the manuscript and made the required changes.

Reviewer comments:

 All the reference is used in the manuscript should be in order form. In this manuscript mostly the reference was used randomly.

Author’s response: We have adapted accordingly to your suggestion. 

Reviewer comments:

Legend in Figures (a) and (b) is not in the same location, it should be in the same location.

Author’s response:

Thank you for highlighting this point. We have corrected it in red.

Reviewer comments:

In addition, the conclusions and abstract must be self-contained.

Author’s response:

We have revised the manuscript and made the correction in red.

Reviewer comments:

 In Methodology Section

The work in the current state must be improved, in the following points:

The presentation of the proposal must be clear and precise. Including models, algorithms, and diagrams.

 Author’s response:

We have revised the manuscript and made the required changes.

Reviewer comments:

A further suggestion for citation, cite the following ready reference in this manuscript

  1. Ullah, R., Gani, A., Shiraz, M., Yousufzai, I.K. and Zaman, K., 2022. Auction Mechanism-Based Sectored Fractional Frequency Reuse for Irregular Geometry Multicellular Networks. Electronics, 11(15), p.2281.

Author’s response:

We thank the reviewer for suggesting the new paper for reference. However, being a very interesting method paper on its own behalf,  after carefully studying the content of the proposed paper, we did not find it relevant to the current discussion in our manuscript. Therefore, we didn’t add this reference to our paper.